# Genome-Wide Analysis of the Almond *AP2/ERF* Superfamily and Its Functional Prediction during Dormancy in Response to Freezing Stress

**DOI:** 10.3390/biology11101520

**Published:** 2022-10-17

**Authors:** Zhenfan Yu, Dongdong Zhang, Shaobo Hu, Xingyue Liu, Bin Zeng, Wenwen Gao, Yawen He, Huanxue Qin, Xintong Ma

**Affiliations:** 1College of Horticulture, Xinjiang Agricultural University, Urumqi 830000, China; 2Guangzhou Institute of Forestry and Landscape Architecture, Guangzhou 510000, China

**Keywords:** *Prunus dulcis*, *AP2/ERF* gene family, evolutionary analyses, freezing stress, expression patterns

## Abstract

**Simple Summary:**

The ethylene-responsive element (AP2/ERF) is one of the key and conserved transcription factors (TFs) in plants, and it plays a crucial role in regulating plant growth, development, and stress response. The cultivated almond in Xinjiang is often affected by short-term ultralow temperature freezing stress during the winter dormancy period, resulting in the death of large-scale almond plants. In this study, we conducted the first genome-wide analysis of the *PdAP2/ERF* family in almond, including protein physicochemical properties, phylogenetic relationships, motif types, gene structures, gene replication types, collinearity relationships, and *cis*-element types in promoter regions. We further analyzed the expression patterns of the *PdAP2/ERF* gene in different tissues of almond and under freezing stress at different temperatures in annual dormant branches using transcriptome data. In addition, we also analyzed the expression levels of 13 *PdAP2/ERF* genes in four tissues of almond and in annual dormant branches treated with freezing stress at different temperatures using fluorescence quantitative technology. This study laid the foundation for further exploring the function of the *PdAP2/ERF* gene in almond.

**Abstract:**

The AP2/ERF transcription factor family is one of the largest transcription factor families in plants and plays an important role in regulating plant growth and development and the response to biotic and abiotic stresses. However, there is no report on the AP2/ERF gene family in almond (*Prunus dulcis*). In this study, a total of 136 *PdAP2/ERF* genes were identified from the almond genome, and their protein physicochemical properties were analyzed. The *PdAP2/ERF* members were divided into five subgroups: AP2, RAV, ERF, DREB, and Soloist. The *PdAP2/ERF* members in each subgroup had conserved motif types and exon/intron numbers. *PdAP2/ERFS* members are distributed on eight chromosomes, with 22 pairs of segmental duplications and 28 pairs of tandem duplications. We further explored the colinear relationship between almond and *Arabidopsis thaliana*, *Oryza sativa*, *Malus domestica*, and *Prunus persica*
*AP2/ERF* genes and their evolution. The results of *cis*-acting elements showed that *PdAP2*/ERF members are widely involved in various processes, such as growth and development, hormone regulation, and stress response. The results based on transcriptome expression patterns showed that *PdAP2/ERF* genes had significant tissue-specific expression characteristics and were involved in the response of annual dormant branches of almond to low-temperature freezing stress. In addition, the fluorescence quantitative relative expression results of 13 representative *PdAP2/ERF* genes in four tissues of ‘Wanfeng’ almond and under six low-temperature freezing treatments of annual dormant branches were consistent with the transcriptome results. It is worth noting that the fluorescence quantitative expression level showed that the *PdERF24* gene was extremely significant at −30 °C, suggesting that this gene may play an important role in the response of almond dormancy to ultralow temperature freezing stress. Finally, we identified 7424 and 6971 target genes based on AP2 and ERF/DREB DNA-binding sites, respectively. The GO and KEGG enrichment results showed that these target genes play important roles in protein function and multiple pathways. In summary, we conducted bioinformatics and expression pattern studies on *PdAP2/ERF* genes, including 13 PdAP2/ERF genes, and performed fluorescence quantitative analysis of annual dormant shoots under different low-temperature freezing stress treatments to understand the tolerance of almond dormancy to freezing stress and suggest future improvements.

## 1. Introduction

Plants often suffer from biotic and abiotic stresses during growth and development, which can cause plant death in severe cases. Therefore, plants have evolved various regulatory mechanisms to resist the harm caused by adverse environmental conditions. Transcription factors, also known as trans-acting factors, are proteins that bind specifically to *cis*-acting elements in eukaryotic promoters [1]. Under different stress conditions, transcription factors regulate stress-responsive genes by specifically binding to *cis*-acting elements in their promoters and activating or suppressing their transcription. For example, a variety of transcription factors, such as *WRKY*, *MyB*, *Mate*, *BZIP*, and *AP2/ERF*, have been identified in research on plant responses to drought, salt-alkali conditions, temperature, and pathogens [2,3].

AP2/ERF (APETALA2/ethylene-responsive element binding factor) is one of the largest transcription factor families in plants. These factors participate in the regulation of growth and development in plants and responses to biotic and abiotic stresses [4]. The protein sequences of the members of this family all contain at least one conserved AP2 domain consisting of approximately 60 amino acids. AP2/ERF members can be divided into three types based on the number of AP2 domains: AP2, RAV (related to ABI3/VP1), and ERF [5]. Among them, AP2 subgroup members contain two AP2 domains and are usually divided into AP2 and AINTEGUMENTA (ANT) monophyletic groups [6]. Members of the ERF subgroup contain only one AP2 domain and can be divided into ten subgroups, A1~A5 and B1~B5, where A1 to A5 are DREB subgroups and B1 to B5 are ERF subgroups. The RAV subgroup members contain an AP2 conserved domain and a B3 DNA-binding motif [7]. Additionally, different accessions with Ap2-like domains but lacking the extra motif are regularly identified as Soloist subgroup members.

Studies of the *AP2/ERF* gene family in plants have yielded many important results. In most cases, AP2 subgroup members are mainly involved in regulating the growth and development of different plant organs, such as epidermal leaf cells, spikelet meristems, grain filling and fruit development [8,9]. The RAV subgroup plays an important role in the transduction of plant hormones (including ethylene and brassinolide) and in the response to biotic/abiotic stress [10]. In addition, members of the DREB and ERF subgroups are mainly affected by biotic and abiotic stresses, such as water deficit, low temperature, and high salt stress [11]. The DREB subgroup is sensitive to low temperature (*AtCBF1*), heat (*ZmDREB2A*, *AtDREB1A*), osmosis (*CkDREB*), drought (*OsDREB1*), and high water stress (*CaDREBLP1*) and shows a clear response pattern under stressful environments [12,13]. *OsDREB1A* and *OsDREB1B* in rice can be induced by cold stress [14], and *OsDREB2A* can be induced by drought and salt stress [15]. Overexpression of *OsDREB1A* enhances the tolerance of transgenic plants to drought, salt, and low-temperature stress [16]. Overexpression of the *JcERF* gene in *Arabidopsis* enhances tolerance to salt and cold stress [17].

Almond (*Prunus dulcis* L.) is one of the most widely planted nut-producing fruit trees in the world, with an annual output of more than 3 million tons worldwide [18]. China’s Xinjiang region is the main planting area of almond, with a planting area of 66,000 hectares. Almond is also an economically important fruit tree in Xinjiang. In the almond planting area of Kashgar, Xinjiang, and the average temperature in winter is −2~8 °C. However, there will also be short-term ultralow temperatures in which the temperature rapidly drops to −20~−30 °C in a short period of time, resulting in large-scale death of almond trees due to freezing damage during the dormant period. Therefore, to fully understand the characteristics of *AP2/ERF* gene family members in almond and determine whether they are involved in the freezing stress response, we mined *AP2/ERF* gene family members from the whole almond genome for gene family and expression analysis. Using bioinformatics methods, we analyzed the protein sequences of *PdAP2/ERF* family members for protein physicochemical properties, phylogenetic tree construction and classification, conserved motifs, gene structure, gene chromosomal location, gene replication type, *cis*-acting elements, and expression patterns. Thirteen *PdAP2/ERF* genes were selected for quantitative expression analysis in leaves, annual branches, flowers, and fruits of the Xinjiang almond cultivar ‘Wanfeng’. These 13 *PdAP2/ERF* genes were also selected for quantitative expression analysis of the annual dormant branches at different temperatures. The purpose of this study was to explore the expression characteristics of *PdAP2/ERF* gene family members in different organs of almond during development and to determine whether these members were involved in the expression response to freezing stress during the dormant period of almond.

## 2. Materials and Methods

### 2.1. Screening of AP2/ERF Gene Family Members

The whole genome data of almond are from the GDR database (https://www.rosaceae.org/analysis/295, accessed on 8 Jul 2022) [19]. The AP2 domain hidden Markov model (PF00847) downloaded from the Pfam database was searched by HMM search based on the whole almond genome protein sequence [20,21]. Protein physicochemical properties were analyzed using the ProtParam online tool (https://web.expasy.org/protparam/, accessed on 8 Jul 2022), and subcellular localization prediction of PdAP2/ERF proteins was performed using BUSCA (http://busca.biocomp.unibo.it, accessed on 8 Jul 2022) [22].

### 2.2. Phylogenetic Tree Construction and Classification

The *Arabidopsis thaliana* AP2/ERF protein sequence was obtained using the TAIR database (https://www.arabidopsis.org/, accessed on 9 Jul 2022). Multiple alignments of AP2/ERF protein sequences in almond and *Arabidopsis thaliana* were performed using the MEGA X tool selection MUSCLE method. In addition, with the Poisson model, complete deletion and 1000 bootstrap replications parameters were selected for neighbor-joining phylogenetic tree construction [23].

### 2.3. Analysis of the Motifs and Gene Structures of PdAP2/ERF Family Members

PdAP2/ERF protein sequence conserved motifs were annotated using the MEME Suite 5.4.1 (https://meme-suite.org/meme/, accessed on 9 Jul 2022) online tool to select default parameters [24]. PdAP2/ERF protein domains were identified using the NCBI CDD database (https://www.ncbi.nlm.nih.gov/cdd, accessed on 9 Jul 2022). The gene structure information of *PdAP2/ERF* members was analyzed based on the whole almond genome Generic Feature Format data. Finally, the TBtools tool was used to draw the cluster diagram [25].

### 2.4. Chromosome Location, Collinearity, and Ka/Ks Value Analysis of the PdAP2/ERF Genes

TBtools was used to extract *PdAP2/ERF* gene information to draw a chromosome distribution map [25]. We used the MCScanX tool to analyze the fragment duplication and tandem duplication types of PdAP2/ERF members [26] and used the Circos tool to draw a genome circle map to display the fragment duplication gene information [27]. In addition, we further selected *AP2/ERF* gene family members from *Prunus dulcis*, *Arabidopsis thaliana*, *Oryza sativa*, *Malus domestica,* and *Prunus persica* to study the collinear gene distribution relationship, and we used the TBtools tool to calculate collinear genes for selective evolutionary pressure analysis (Ka, Ks, and Ka/Ks).

### 2.5. Analysis of cis-Elements in PdAP2/ERF Genes

We extracted the 2000-bp promoter sequence upstream of each *PdAP2/ERF* gene from the almond genome data, and *cis*-acting element predictions were performed on the PlantCARE online website [28].

### 2.6. Expression Pattern Analysis

Transcriptome data from four almond leaf tissues (SRR12806975) [29], flowers (SRR16267429) [30], flower buds (SRR11251344) [31], and fruitlets (SRR10189226) [32] were downloaded from the NCBI SRA database to obtain the corresponding fragments per kilobase of exon model per million mapped fragments (FPKM) value, and the expression patterns of *PdAP2/ERF* genes in different tissues were analyzed. We analyzed the expression patterns of *PdAP2/ERF* gene members using transcriptome data of one-year dormant branches of almond treated with freezing stress (−5 (CK), −10, −15, −20, and −25 °C) that were obtained by our sequencing team (GSA Number: CRA007323).

### 2.7. Material Handling and qRT–PCR Analysis

To explore the changes in *AP2/ERF* gene expression levels in the annual dormant branches of almond subjected to freezing stress, we selected 6-year-old ‘Wanfeng’ almond from the almond resource garden in Shache County, Xinjiang, in January 2022. We selected three trees with robust and similar growth and selected annual dormant branches with similar growth from the middle of the canopy. The notch was sealed with paraffin and taken back to the laboratory for processing. One dormant branch was taken from three trees as three biological replicates, and sets of three dormant branches were used as a treatment group. Finally, freezing stress treatment was performed at six temperatures (−5 (CK), −10, −15, −20, −25, and −30 °C), and materials were stored in liquid nitrogen for future use. In addition, in May 2022, we used the same method to select leaves, annual branches, fruit pulp, and fruit cores from the six-year-old ‘Wanfeng’ almond trees, with three biological replicates for each tissue; samples were stored at −80 °C for future use.

We selected 13 representative *PdAP2/ERF* genes for qRT–PCR analysis. Three independent biological repeats containing three independent plants were used for qRT–PCR detection. Total RNA was extracted using an Phygene plant RNA extraction kit (PH0233); a reverse transcription reaction kit (RR047Q) produced by Baori Doctor Biotechnology (Beijing, China) Co., Ltd., was used for the reverse process; for quantitative qRT–PCR, the chimeric fluorescence detection enzyme (quantitative kit) (RR820Q) produced by Baori Doctor Biotechnology (Beijing, China) Co., Ltd., was used. Quantitative primers were designed using Primer 5 software, and the primers were synthesized by Wuhan Transduction Biology Laboratory Co., Ltd. (Wuhan, China) (Appendix A). qRT–PCR quantification was performed using a YT-PCR2 fluorescence quantitative PCR instrument, and the reaction program was set as follows: 95 °C for 30 s, 95 °C for 5 s, 60 °C for 30 s, and 40 cycles. The obtained cycle threshold (CT) values were quantitatively analyzed by the 2^−ΔΔCT^ method (Appendix A) [33].

### 2.8. Construction of the PdAP2/ERF Protein Interaction Network

PdAP2/ERF protein sequences were uploaded to the STRING database for protein interaction network prediction (Appendix A) [22]. Gene Ontology (GO) annotation of *PdAP2/ERF* genes was performed using Tbtools (Appendix A).

### 2.9. Identification and Annotation of Target Genes of PdAP2/ERF Genes

Based on the protein interaction prediction results, we downloaded data on the AP2 DNA-binding site (MA0571.1) (Appendix A) and ERF/DREB DNA-binding site (MA0976.2) (Appendix A) from the JASPA_CORE database (http://jaspar.genereg.net, accessed on 13 Jul 2022) [34]. To obtain the potential downstream target genes regulated by *AP2* and *ERF/DREB*, the upstream 2000-bp promoter sequence of each almond gene was extracted by TBtools. Then, the Motif FIMO (https://meme-suite.org/meme/, accessed on 13 Jul 2022) tool was used to detect the motifs in the almond promoters that bound to *AP2* and *ERF/DREB* family members. Target genes of candidate *AP2* and *ERF/DREB* family members were functionally annotated using GO and the Kyoto Encyclopedia of Genes and Genomes (KEGG) database.

## 3. Results

### 3.1. Analysis of the Characteristics of PdAP2/ERF Family Members

A total of 136 *AP2/ERF* genes were screened from the whole genome of almond, and each protein sequence contained at least one AP2 domain. Based on the results of protein homology alignment with *Arabidopsis AP2/ERF* members, the *PdAP2/ERF* genes were renamed according to the order on the 8 almond chromosomes. They are named *PdAP2-1* to *PdAP2-19*, *PdRAV1* to *PdRAV6*, and *PdERF1* to *PdERF98*. In addition, 11 pairs of homologous genes were renamed *PdAP2-4a/4b*, *PdAP2-8a/8b*, *PdAP2-11a/11b*, *PdAP2-13a/13b/13c/13d*, *PdAP2-19a/19b*, *PdERF2a/2b*, *PdERF33a/33b*, *PdERF38a/38b*, *PdERF70a/70b*, *PdERF74a/74b*, and *PdERF89a/89b*. The physicochemical properties of *PdAP2/ERF* family members were analyzed (Appendix A). Among them, the length of the protein sequence ranged from 99 aa (PdERF16) to 862 aa (PdAP2-16), and the molecular weight of the protein ranged from 11.913 kDa (PdERF16) to 93.975 kDa (PDAP2-16). The isoelectric points ranged from 4.31 (PdERF46) to 11.57 (PdERF46), and 53 (PI > 7) and 83 (PI < 7) PdAP2/ERF members were basic and acidic proteins, respectively. The GRAVY (grand average of hydropathicity) value for PdAP2/ERF member proteins was less than 0, indicating that they were hydrophilic. The subcellular prediction of PdAP2/ERF proteins showed that 120, 15, and 1 were in the nucleus, chloroplast, and cytoplasm, respectively (Appendix A).

### 3.2. Multiple Sequence Alignments and Phylogenetic Relationships of the PdAP2/ERF Family Members

To explore the phylogenetic and taxonomic relationships of PdAP2/ERF proteins, we constructed two phylogenetic trees of *PdAP2/ERF* members using neighbor-joining methods. According to the branch clustering results of the phylogenetic tree, the *PdAP2/ERF* members were clearly clustered into four subgroups, *AP2*, *RAV*, *ERF*, and Soloist, with 26, 6, 103, and 1 member, respectively (Figure 1). Based on the *Arabidopsis thaliana* AP2/ERF classification results, *PdERF* members were further divided into two subgroups, DREB (A1~A6) and *ERF* (B1~B6). Among them, DREB contains four groups, I, II, III, and IV, and ERF contains eight groups, V, VI, VII, VIII, IX, X, Xb-L, and VI-L. In addition, we further compared the number of *AP2/ERF* members in almond with those in *Arabidopsis thaliana*, *Oryza sativa*, *Malus domestica*, and *Prunus persica* (Appendix A) and found that the number of *AP2/ERF* members in almond and peach was similar across different subgroups and groups, indicating that the two plants have a low degree of evolutionary separation and a high degree of conservation.

### 3.3. Motif and Gene Structure Compositions of PdAP2/ERF Family Members

We constructed a neighbor-joining phylogenetic tree of PdAP2/ERF gene members (Figure 2A). Ten motifs in 136 PdAP2/ERF proteins were annotated by the MEME online tool and clustered using phylogenetic trees (Figure 2B). The results showed that the *PdAP2/ERF* members of the same subgroup had relatively conserved motif types. There are four main types of motifs in the *AP2* subgroup: motifs 2, 3, 5, and 6. There are three main types of motifs in the RAV subgroup: motifs 2, 3, and 4. There are five main types of motifs in the *DREB* and *ERF* subgroups: motifs 1, 3, 4, 7, and 8. Motifs 1, 2, 3, 4, and 7 are all *AP2* types and are all located within the *AP2* domain region (Figure 2C,E). Furthermore, all members of the RAV, DREB, and ERF subgroups have only one AP2 domain. In the AP2 subgroup, all members have two *AP2* domains, except for PdAP2-16, which has only one AP2 domain (Figure 2C).

The number of exons and introns of the 136 *PdAP2/ERF* members was counted and clustered using the phylogenetic tree (Figure 2D). The results showed that members in the same subgroup had similar numbers of exons and introns, and the location distribution was highly conserved. Among them, the AP2 subgroup has 6~10 exons and 5~9 introns. *PdERF93* in the Soloist subgroup has 6 exons and 5 introns. In the RAV, DREB, and ERF subgroups, 5, 37, and 48 genes have only 1 exon and 0 introns, respectively, while the remaining 19 genes have 2~3 exons and 1~2 introns.

### 3.4. Gene Mapping, Collinearity, and Ka/Ks Values of PdAP2/ERF Family Members

According to the distribution and location of genes in the almond genome, the chromosomal locations of *PdAP2/ERF* genes were analyzed. In addition, we further set the genetic spacing to 200 kb to calculate the gene density of each chromosome (Appendix A), which is represented by a gradient color from blue (low gene density) to red (high gene density) in Figure 3. Empty regions on chromosomes indicate genetic regions lacking gene distribution information. As shown in Figure 3, 136 *PdAP2/ERF* genes were unevenly distributed across 8 chromosomes, and Pd01~Pd08 had 28, 18, 16, 6, 25, 20, 15, and 8 genes, respectively. It is worth noting that most of the *PdAP2/ERF* genes are mainly distributed in regions with a high gene density. Similar to other studies on plant *AP2/ERF* families, there was no significant correlation between chromosome length and the number of *PdAP2/ERF* genes.

We further performed collinearity analysis on *PdAP2/ERF* family members to explore the type of gene duplication. A total of 22 pairs of segmental duplication genes and 28 pairs of tandem duplication genes were identified in the *PdAP2/ERF* family members (Appendix A). Segmental duplication and tandem duplication events occurred on all 8 chromosomes. Pd01~Pd08 have 3, 6, 2, 1, 9, 3, 3, and 1 pairs of tandem repeat genes (Figure 3). We further drew a Circos diagram of the distribution of segmental duplication genes within chromosomes (Figure 4). There were 8, 6, 5, 1, 5, 5, 3, and 2 segmental duplication genes on Pd01~Pd08. Notably, most of the duplicated genes were from two subgroups, DREB and ERF.

To explore the evolutionary history of the members of the *PdAP2/ERF* family in almond, we constructed a collinear map of the *AP2/ERF* gene members in almond and those in four species: *Arabidopsis thaliana*, *Oryza sativa*, *Malus domestica,* and *Prunus persica* (Figure 5). The results showed that there were 56, 26, 164, and 109 pairs of collinear *AP2/ERF* genes between almond and *Arabidopsis thaliana*, *Oryza sativa*, *Malus domestica* and *Prunus persica*, respectively (Appendix A). It is worth noting that some *PdAP2/ERFs* have a collinear relationship with two or more *AP2/ERF* members in *Arabidopsis thaliana*, *Oryza sativa,* and *Malus domestica*, suggesting that these genes play important roles in the evolution of the *PdAP2/ERF* gene family. There is a one-to-one collinearity in genes from almond and peach. It is speculated that the *AP2/ERF* genes of these two plants have a low degree of separation during evolution and are highly conserved.

We further calculated the Ka, Ks, and Ka/Ks values of segmental duplication, tandem duplication, and interspecies collinear genes to evaluate the selection on *PdAP2/ERF* family members during evolution. The results showed that the segmental duplication, tandem duplication, and interspecies collinear gene Ka/Ks values among *PdAP2/ERF* family members were all less than 1 (Appendix A). Therefore, we speculate that members of the almond *PdAP2/ERF* family may have undergone strong purifying selection during evolution.

### 3.5. Upstream cis-Regulatory Elements

A total of 123 *cis*-acting elements were annotated in the 2000-bp upstream promoter region of 136 *PdAP2/ERF* genes, and 74 elements had well-defined functions (Appendix A). The results showed that in addition to CAAT-box and TATA-box, there are also many light-related elements, such as ACE, G-box, and P-box, in the promoter region. In addition, we further mapped the types and locations of hormone regulation-, stress response-, and growth- and development-related elements (Figure 6; Appendix A). Among them, hormone regulatory elements include auxin (AuxRE, AuxRR-Core, and TGA-Box), gibberellin (TATC-box, P-box, and GARE-motif), salicylic acid (SARE), and methyl jasmonate (TGACG-motif). The stress response elements include five types: ARE, LTR, TC-rich repeats, MBS, and WUN-motif. The growth and development elements include eight types: CAT-box, O2-site, RY-element, circadian, HD-Zip 1, MSA-like, AACA-motif, and motif I. In conclusion, the promoter element results suggest that *PdAP2/ERF* members are widely involved in the growth and development of almond.

### 3.6. Expression Pattern Analysis of the PdAP2/ERF Family Members

Genes with FPKM values less than 1 were removed from datasets generated for the leaves, flowers, flower buds, and fruitlets of almond, and 96 *PdAP2/ERF* genes were finally retained for tissue expression pattern analysis (Appendix A). As shown in Figure 7A, 16, 13, 37, and 28 *PdAP2/ERF* genes were significantly upregulated in leaves, flowers, flower buds, and fruitlets, respectively. This result indicated that the *PdAP2/ERF* genes had significant tissue-specific expression characteristics. It is worth noting that three genes, *PdERF29*, *PdERF34*, and *PdERF78,* were highly expressed in leaves; *PdERF68* was highly expressed in flowers; *PdERF7* was highly expressed in flower buds; and *PdERF13*, *PdERF38b*, *PdERF48*, *PdERF51*, *PdERF54*, *PdERF88*, and *PdERF92* were highly expressed in fruitlets. These results suggest that these genes may play an important role in the growth and development of the corresponding tissue in almond.

We further examined the transcriptome data of dormant annual branches treated with freezing stress. After removing genes with FPKM values less than 1, we finally obtained 80 *PdAP2/ERF* genes to analyze the expression patterns of dormant one-year branches under freezing stress at different temperatures (Appendix A). As shown in Figure 7B, 14, 6, 12, 6, and 27 genes were significantly upregulated at −5 (CK), −10, −15, −20, and −25 °C, respectively. It is worth noting that AP2 subgroup members are mainly expressed at −15 and −20 °C, RAV subgroup members are mainly expressed at −10 °C, and ERF and DREB subgroup members are mainly expressed at −25 °C. In conclusion, it is speculated that *PdAP2/ERF* family members are widely involved in the response of annual dormant almond branches to freezing stress, and ERF and DREB subgroup members may play a major role in the response to ultralow temperature stress.

### 3.7. PdAP2/ERF Gene Quantitative qRT–PCR Analysis

We used qRT–PCR to analyze the expression changes of 13 *PdAP2/ERF* genes in four tissues of ‘Wanfeng’ almond: leaves, branches, flesh, and cores. According to the qRT–PCR results, 13 *PdAP2/ERF* genes had significant tissue-specific expression characteristics (Figure 8). We found that the relative expression levels of eight genes, *PdERF21*, *PdERF24*, *PdERF35*, *PdERF39*, *PdERF44*, *PdERF48*, *PdERF51*, and *PdAP2-6,* were significantly increased in branches. The relative expression levels of three genes, *PdERF13*, *PdERF31*, and *PdERF47*, were significantly increased in flesh, and the relative expression levels in the cores were also higher. This result indicated that the *PdAP2/ERF* genes may be involved in the growth and development of different tissues of almond.

Similarly, the expression changes of 13 *PdAP2/ERF* genes in annual dormant branches treated with freezing stress at six temperatures were measured by qRT–PCR (Figure 9). According to the qRT–PCR results, eleven genes, *PdERF3*, *PdERF21*, *PdERF24*, *PdERF31*, *PdERF35, PdERF39, PdERF47, PdERF48, PdERF51, PdERF53*, and *PdAP2-6,* showed a relative increase in their expression levels in annual dormant branches under freezing stress at different temperatures. It is worth noting that the relative expression level of the *PdERF24* gene was 324 times higher than that of the control under freezing stress at −30 °C. Under −25 °C treatment, the relative expression levels of three genes, *PdERF3*, *PdERF21* and *PdERF39,* were 12, 15, and 30 times that of the control, respectively. However, the relative expression levels of *PdERF35*, *PdERF47*, *PdERF48*, *PdERF51*, and *PdERF53* showed a similar increase under freezing stress at different temperatures. This indicated that *PdAP2/ERF* genes may be involved in the effect of almond dormancy under low-temperature freezing stress.

### 3.8. Protein Interactions and GO Annotations of PdAP2/ERF Members

We predicted potential protein interactions among PdAP2/ERF members using the STRING database. There were 50 nodes in the PdAP2/ERF protein interaction network, and not all nodes interacted with each other (Figure 10). There were 40 *PdERF* and *10 PdAP2* genes involved in the protein interaction network. Some proteins showed direct interactions, such as *PdERF39* and *PdAP2-14*, while other proteins exhibited more complex multiprotein interactions, such as *PdAP2-5*, *PdERF8*, and *PdERF63*. In addition, *PdAP2-14* and *PdERF93* were predicted to be central nodes, and there were protein interactions among 16 and 14 genes, respectively. Notably, no protein interactions were predicted for the *PdRAV* genes.

We further performed GO annotation and enrichment analysis on 50 PdAP2/ERF proteins. The results showed that a total of 18 PdAP2/ERF proteins were annotated to GO functions and distributed in the molecular function and biological process categories. The top 20 significant items were statistically enriched based on *p* value. In the Molecular Function category, nucleic acid binding transcription factor activity, transcription factor activity, sequence-specific DNA binding, DNA binding, and nucleic acid binding were significant (Figure 11A). In the biological process category, mainly metabolism- and biosynthesis-related items, such as regulation of RNA metabolic process and regulation of cellular biosynthetic process, were significant (Figure 11B).

### 3.9. Identification and Annotation of PdAP2/ERF Target Genes

To identify the possible downstream target genes regulated by *PdAP2/ERF*, we searched the 2000-bp upstream promoter sequences of the almond genes using the conserved AP2 and ERF/DREB motifs in the JASPAR database. The results showed that 7424 target genes were found for AP2, 1150 target genes had two types of binding sequences (Appendix A), and GO annotations were obtained for 2260 target genes (Appendix A). According to the enrichment results (Appendix A), the target genes in the Biological Process category were mainly distributed in cellular process (GO:0009987) and metabolic process (GO:0008152), followed by single-organism process (GO:0044699). The target genes in the cellular component category were mainly distributed in cell (GO:0005623) and cell part (GO:0044464), followed by organelle (GO:0043226) and membrane (GO:0016020). Target genes in the Molecular Function category are mainly distributed in binding (GO:0005488) and catalytic activity (GO:0003824). According to the results of the top 20 GO terms with significant enrichment (Figure 12; Appendix A), target genes were mainly enriched in metabolism-related GO terms such as regulation of metabolic process (GO: 0019222) and macromolecule metabolic process (GO: 0044260). Similarly, 3414 target genes were enriched in multiple KEGG pathways (Appendix A; Appendix A). Among them, a total of 2492 target genes were enriched in metabolism-related pathways. According to the KEGG pathways with the most enriched target genes, metabolic pathways (KO01100), biosynthesis of secondary metabolites (KO01110), and starch and sucrose were included in metabolism (KO00500) and other pathways.

A total of 6971 target genes were found for ERF/DREB. Different from the AP2 target gene-binding site sequence, there are two or more types of binding sequences in many genes, and these are shorter than the AP2 target gene binding site sequence. There were 1593 identical target genes (Appendix A). A total of 2471 target genes had GO annotations (Appendix A). The GO enrichment results of the AP2 target genes in the biological process, cellular component, and molecular function categories were similar (Appendix A). Moreover, the top 20 GO terms with significant enrichment of ERF/DREB and AP2 target genes were similar (Figure 12; Appendix A). The metabolic process of tRNA (GO:0006399), NADP biosynthetic process (GO:0006739), and hydrogen ion transmembrane were commonly found. A variety of metabolism-related GO terms, such as transport (GO:1902600), were significantly enriched. Similarly, the KEGG metabolic pathways enriched by ERF/DREB and AP2 target genes were basically consistent, and a total of 2238 target genes were enriched in metabolism-related pathways (Appendix A; Appendix A). In summary, AP2 and ERF/DREB can affect multiple pathways by regulating downstream target genes, and the two have large differences in the number of target genes and the length of the binding site sequence. The GO and KEGG enrichment results were highly similar. Therefore, we speculate that *AP2* and *ERF/DREB* members in almond have important complementary roles in regulating downstream target genes, thereby responding to changes in various environmental conditions.

## 4. Discussion

There are various unfavorable factors in nature that affect the growth and development of plants. Therefore, plants have developed a series of response mechanisms, among which transcription factors play an important role. AP2/ERF transcription factors are ubiquitous in plants, and these transcription factors play an important role in plant growth and development and in response to environmental stress [4]. The *AP2/ERF* gene family has been extensively reported in many plants, such as *Arabidopsis thaliana* [35], *Saccharum spontaneum* [36], *Triticum durum* [37], wheat [38], maize [39], rubber tree [40], and Populus [41]. However, no research has been reported on the almond *AP2/ERF* family.

The number of *PdAP2/ERF* genes in almond is similar to that in peach [42]. In addition, based on previous research results, there was no absolute relationship between the genome size of a species and the number of *AP2/ERF* gene family members. Based on the Arabidopsis *AP2/ERF* subgroup classification, the *PdAP2/ERF* family members were clearly divided into four subgroups, *AP2*, *RAV, ERF*, and *Soloist1*, and *ERF* was further divided into two subgroups, *ERF* and *DREB*. It is worth noting that the number of *AP2/ERF* family members in the subgroups and groups of almond and peach was similar, indicating that the two have a low degree of separation and high conservation.

According to related studies, gene functions and conserved domains in amino acid sequences are closely related. Conserved motifs in AP2/ERF transcription factors are critical for their functions, such as transcriptional activity [35]. A total of 10 motifs were annotated to *PdAP2/ERF* family members, and members belonging to the same subgroup had more similar motif types. This means that *PdAP2/ERF* genes within the same subgroup may be highly conserved during evolution. Gene structure (intron–exon) information can provide evolutionary clues for family members [43]. The *PdAP2/ERF* genes in the AP2 subgroup have more exons and introns, and the number of exons and introns in the three subgroups RAV, ERF, and DREB is more conserved. These results are consistent with the results of the gene structure of peach *AP2/ERF* family members, and the two AP2 conserved domain sequences found in both species were both broken into multiple segmentals by introns [42].

Gene duplication is the main factor leading to the rapid expansion and evolution of gene families, and it is also an important means to help plants cope with environmental changes during growth and development [44]. At present, gene duplication events of *AP2/ERF* family members have been studied in a variety of plants. However, the gene duplication events of almond *AP2/ERF* family members have not been studied. In this study, 22 pairs of segmental duplications and 28 pairs of tandem repeat duplications were identified in *PdAP2/ERF* family members. The number of duplicated *AP2/ERF* genes in almond was lower than that in *Prunus mume* and apple [45,46]. *PdAP2/ERF* family members were unevenly distributed on eight chromosomes, among which there were at least 6 genes on chromosome Pd04, while there were at most 28 genes on chromosome Pd01. We speculated that this uneven distribution might be caused by gene duplication. The uneven distribution of *PdAP2/ERF* family members across the eight chromosomes can also be explained by the number of segment duplications and tandem duplications present on different chromosomes. In addition, *AP2/ERF* family members in almond and *Arabidopsis thaliana*, *Oryza sativa*, *Malus domestica*, and *Prunus persica* shared 56, 26, 164, and 109 pairs of collinear genes, respectively, indicating that the *AP2/ERF* genes in Rosaceae were more conserved during evolution and had a high degree of separation from those in monocots. Notably, some *PdAP2/ERF* genes have a collinear relationship with two or more *AP2/ERF* genes in *Arabidopsis thaliana*, *Oryza sativa*, and *Malus domestica*, suggesting that these genes play an important role in the evolution of the *PdAP2/ERF* gene family. Exploring the Ka/Ks ratios of duplicated genes is an effective way to study the effects of duplicated genes on evolution. In the PdAP2/ERF segmental and tandem repeat gene pairs in which values were calculated, the Ka/Ks values of duplicate genes were all less than 1, indicating that these genes mainly underwent purifying selection. Similarly, the results of collinear genes among different species showed that the Ka/Ks values of the collinear genes were all less than 1, indicating that the evolution of *AP2/ERF* in different species was also under strong purifying selection pressure. The results of *cis*-acting elements showed that *PdAP2/ERF* genes were involved in various functions, such as almond growth and development, stress response, and hormone regulation.

Functional annotation of *AP2/ERF* genes indicated that stress-related *AP2/ERF* genes were enriched in multiple copies in *Arabidopsis thaliana*, and the differential retention rate of repeated sequences in the same transcription factor family was higher [35]. The subgroups involved in the environmental response may help plants adapt to changing environmental conditions [47]. We used different RNA-seq data from almond to analyze the expression patterns of *PdAP2/ERF* family members in different tissues. The results showed that 16, 13, 37, and 28 *PdAP2/ERF* genes were upregulated in leaves, flowers, buds, and young fruits, respectively. This result indicated that the *PdAP2/ERF* family members had remarkable tissue-specific expression characteristics. In the DREB subgroup, a specific structure, the cold-inducible element (A/GCCGAC), may function in the response to abiotic stresses such as low temperature [48]. In *Arabidopsis*, there are three cold-inducible *CBF* genes, CBF1-3 (*CBF1*/*DREB1B*, *CBF2*/*DREB1C*, and *CBF3*/*DREB1A*), which play an important role in regulating the low-temperature response [49]. We further analyzed the expression patterns of *PdAP2/ERF* family members under freezing stress at different temperatures. A total of 65 genes were significantly upregulated at different low temperatures. Among them, the expression of *PdERF63* increased 22 times compared with CK (−5 °C) after freezing stress at −25 °C. The expression levels of *PdERF80*, *PdERF81*, *PdERF91*, and *PdERF98* were also significantly increased. It is speculated that these genes play an important role in the response of almond to freezing stress. In this study, it was found that *PdERF*-type genes in almond were the most significant genes in response to low temperature and showed similar results to those in alfalfa [50]. According to previous research results, ERF-type genes have higher expression levels under stress than AP2-type genes, which may be due to the fewer introns and faster response speed of *ERF* members. Therefore, we speculate that the same phenomenon exists in *PdAP2/ERF*.

The *AP2/ERF* gene family has been intensively studied in a variety of plants. At present, *AP2/ERF* gene family studies have found that *DREB* members are most closely related to chilling stress [48]. Various studies have found that overexpression of homologous *DREB* genes from rice, maize, and wheat in transgenic tobacco and Arabidopsis can improve their cold tolerance [51,52]. In addition, significant research achievements have been made in crop yield research. It has been identified in rice that an *OsDREB1C* can drive a transcription program with diverse functions, determining photosynthetic capacity, nitrogen utilization, and flowering time. The yield of rice overexpressing *OsDREB1C* has increased by 41.3%~68.3% [53]. In this study, we preliminarily explored the relative expression levels of 13 representative *PdAP2/ERF* genes in four tissues of ‘Wanfeng’ almond and six freezing stress temperature treatments in annual dormant branches based on qRT–PCR technology. The leaves, branches, flesh, and cores of the four almond tissues have significant tissue-specific expression characteristics, indicating that *PdAP2/ERF* gene members are involved in the tissue-specific growth and development of almond. In addition, the relative expression levels of 13 *PdAP2/ERF* genes were significantly different in the annual dormant branches of ‘Wanfeng’ almond under different freezing temperatures. Among them, 11 *PdAP2/ERF* genes showed increased relative expression levels at corresponding temperatures after the annual dormant branches were subjected to freezing stress. Therefore, we speculate that *PdAP2/ERF* genes are involved in the effect of almond dormancy under freezing stress and improve the tolerance of dormant almond branches to freezing stress. We found that the expression level of *PdERF24* gene increased in varying degrees under four freezing stresses of −10, −15, −20, and −25 °C. Compared with CK (−5 °C) and −25 °C, the expression level of *PdERF24* gene under −30 °C freezing stress increased 324 and 40 times respectively, reaching 324. It is worth noting that the expression of the *PdERF24* gene was also significant in normal growth and development branches. We hypothesized that *PdERF24* gene may be an important comprehensive regulatory gene in the growth and development of almond branches, and it is worth further investigating the specific function of this gene.

We predicted potential protein interactions among *PdAP2/ERF* members. Fifty PdAP2/ERF members formed a protein interaction network with 70 nodes. Among them, PdAP2-14 and PdERF93 were predicted to be central nodes. *AP2/ERF* genes have been shown to be downstream genes of a series of transcriptional regulatory networks [54,55]. However, to date, few studies have been conducted on the regulation of downstream target genes of *AP2/ERF* genes, which are widely involved in multiple response mechanisms in plants. Therefore, we selected the target genes of *AP2* and ERF/DREB in almond for preliminary research. An *EARLY BUD-BREAK* gene was identified in Japanese pear, which may be an AP2/ERF target gene that regulates the process of flower bud germination [56]. A total of 7424 and 6971 target genes were identified based on AP2 and ERF/DREB DNA-binding sites, respectively, both of which had 1593 identical target genes and similar GO and KEGG enrichment results. Significant GO enrichment was found in metabolism-related terms, and KEGG enrichment showed that most target genes were enriched in metabolism-related pathways. In addition, we found a difference between the two subgroups. *AP2* and *ERF/DREB* differ greatly in the number of target genes and the length of the binding site sequence. Therefore, we speculate that *AP2* and *ERF/DREB* members have important complementary roles in regulating downstream target genes, thereby responding to changes in various environmental conditions.

## 5. Conclusions

In this study, 136 *PdAP2/ERF* genes were identified from the almond genome, and the characteristics of *PdAP2/ERF* family members, such as phylogenetic tree clustering, motifs, gene structures, collinearity, evolution, and upstream *cis*-elements, were comprehensively explored by bioinformatics technology. The expression levels of 13 *PdAP2/ERF* genes in four groups of ‘Wanfeng’ almond leaves, branches, flesh, and cores were investigated by fluorescence quantitative technology. In addition, we also used fluorescence quantitative technology to explore the expression levels of 13 *PdAP2/ERF* genes in the dormant annual branches of ‘Wanfeng’ almond. In conclusion, this study provides a reference for our subsequent in-depth study of the *PdAP2/ERF* gene in improving the resistance of almond to freezing stress.

## Figures and Tables

**Figure 1 biology-11-01520-f001:**
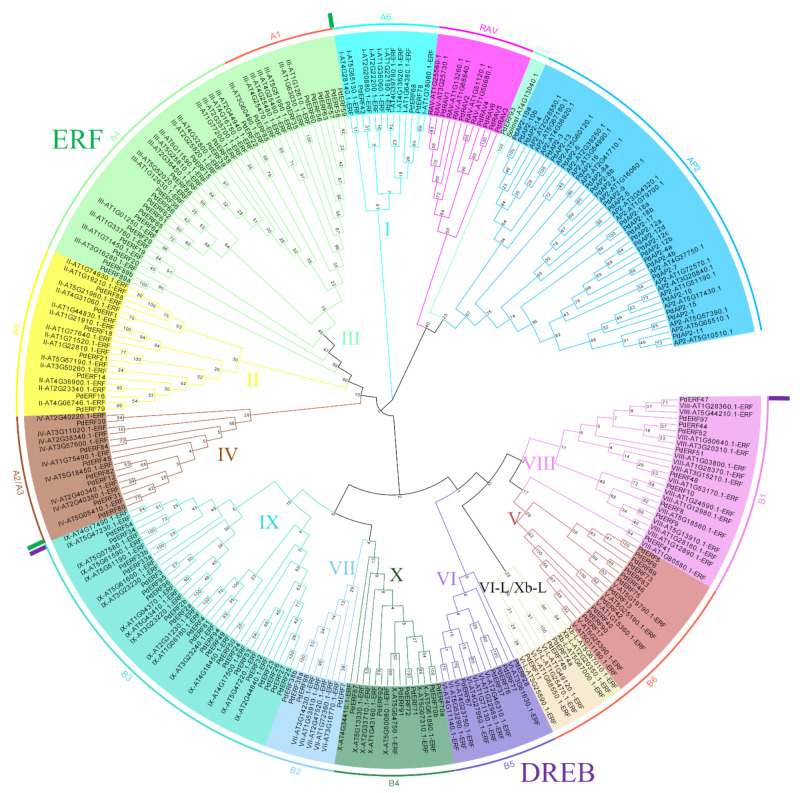
Phylogenetic tree of *AP2/ERF* family members in almond and *Arabidopsis*. Each color block represents a group. The area between the two green segments represents the ERF subgroup; the area between two purple segments represents DREB subgroup.

**Figure 2 biology-11-01520-f002:**
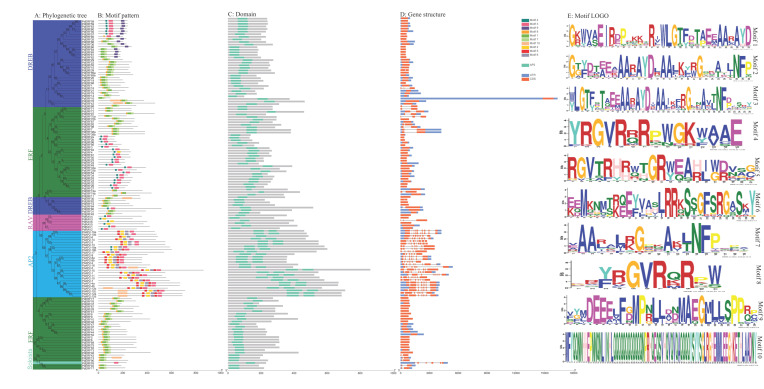
Phylogenetic clustering of *PdAP2/ERF* members based on motifs and gene structures. (**A**) Neighbor-joining phylogenetic tree of members of the *PdAP2/ERF* family. Different colored areas represent different groups. (**B**) Conserved protein motifs. (**C**) AP2 domains. (**D**) Gene structures. (**E**) Motif LOGO. The ruler at the bottom of B and C figures represents the length of amino acids. The ruler at the bottom of D figure represents the number of nucleotides (base pair: bp).

**Figure 3 biology-11-01520-f003:**
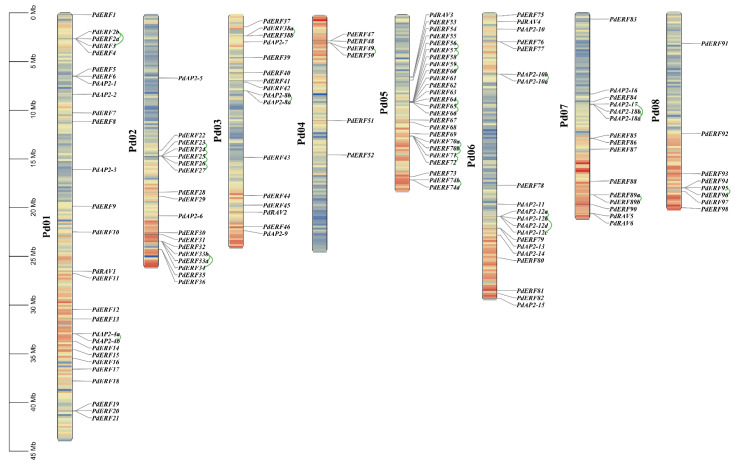
Chromosome localization of almond *PdAP2/ERF* family members. The green lines show tandem duplicated genes. The scale on the left shows the chromosome length information, and the chromosome names are listed on the right. The blue regions in the chromosome segments represent low gene density, and the red regions represent high gene density.

**Figure 4 biology-11-01520-f004:**
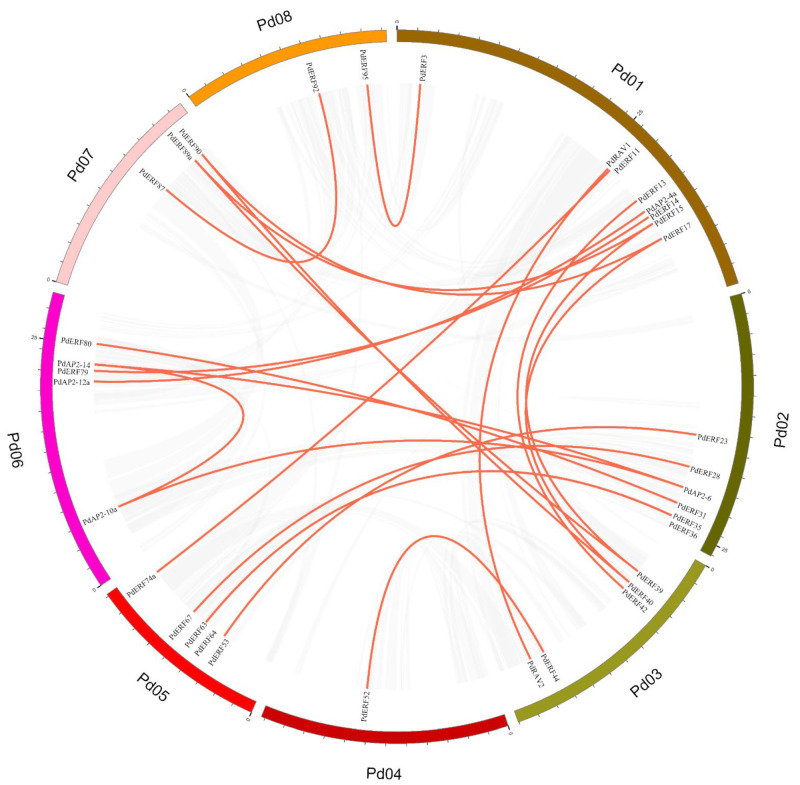
Interchromosomal relationships of the *PdAP2/ERF* genes in the almond genome. The red lines link the segmentally duplicated *PdAP2/ERF* gene pairs.

**Figure 5 biology-11-01520-f005:**
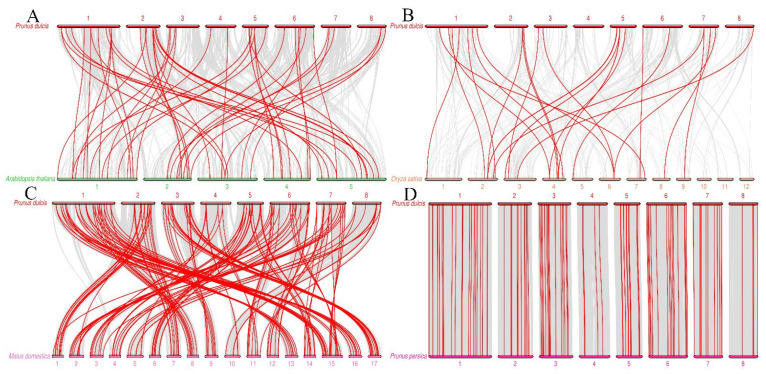
Synteny analysis of *AP2/ERF* family members in almond compared to those from four other plant species. (**A**) *Prunus dulcis* vs. *Arabidopsis thaliana*, (**B**) *Prunus dulcis* vs. *Oryza sativa*, (**C**) *Prunus dulcis* vs. *Malus domestica*, and (**D**) *Prunus dulcis* vs. *Prunus persica*. Each horizontal line represents a chromosome, and the red lines represent collinear genes. Eight chromosomes of *Prunus dulcis* are represented by red segments. Five chromosomes of *Arabidopsis thaliana* are represented by green segments. Twelve chromosomes of *Oryza sativa* are represented by yellow segments. Seventeen chromosomes of *Malus domestica* are represented by light pink segments. Eight chromosomes of *Prunus persica* are represented by pink fragments. The number represents the corresponding chromosome name.

**Figure 6 biology-11-01520-f006:**
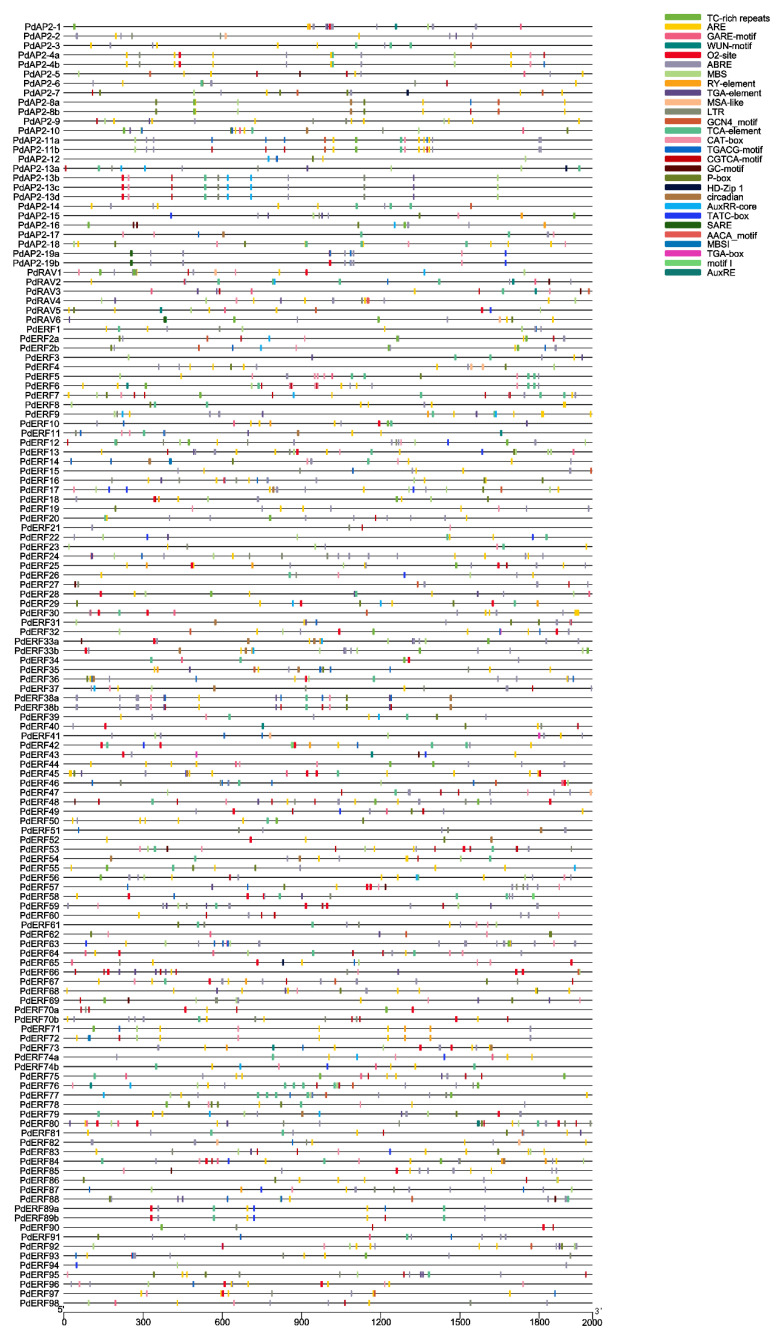
*Cis*-acting elements in the promoter regions of *PdAP2/ERF* family members. Each color square represents a *cis*-element type.

**Figure 7 biology-11-01520-f007:**
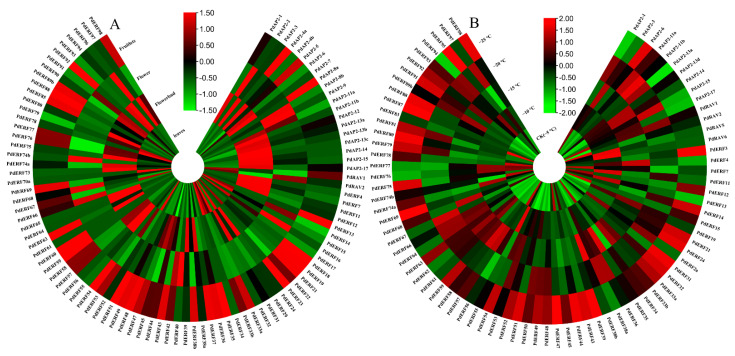
Heatmap of the expression patterns of *PdAP2/ERF* gene family members. (**A**): Heatmap of expression patterns in different tissues. (**B**): Expression heatmap of annual dormant branches under freezing stress at different temperatures. The ROW normalization method was used to draw the heatmap. Red squares indicate upregulation of expression, black squares indicate no expression, and green squares indicate downregulated expression.

**Figure 8 biology-11-01520-f008:**
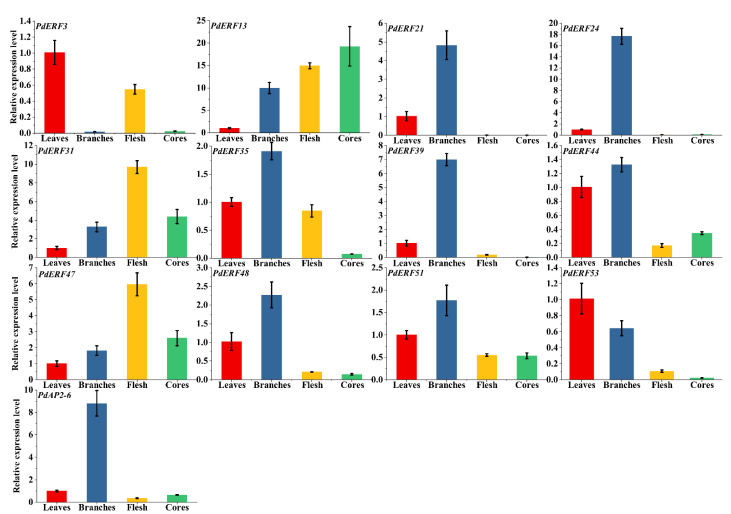
Fluorescence quantitative relative expression of 13 representative *PdAP2/ERF* genes in leaves, branches, flesh, and cores of ‘Wanfeng’ almond. The ordinate represents the relative expression level of genes, and the abscissa represents leaves, branches, flesh, and cores from left to right.

**Figure 9 biology-11-01520-f009:**
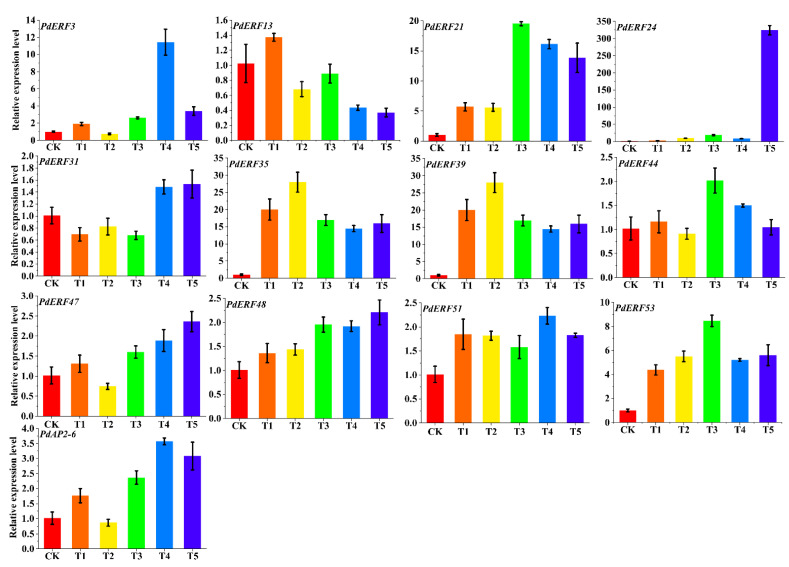
Fluorescence quantitative relative expression of 13 representative *PdAP2/ERF* genes under six temperature gradient freezing stress treatments in the annual dormant branches of ‘Wanfeng’ almond. CK, T1, T2, T3, T4, and T5 represent −5, −10, −15, −20, −25, and −30 °C, respectively. The ordinate represents the relative expression level of genes, and the abscissa represents CK, T1, T2, T3, T4, and T5 from left to right.

**Figure 10 biology-11-01520-f010:**
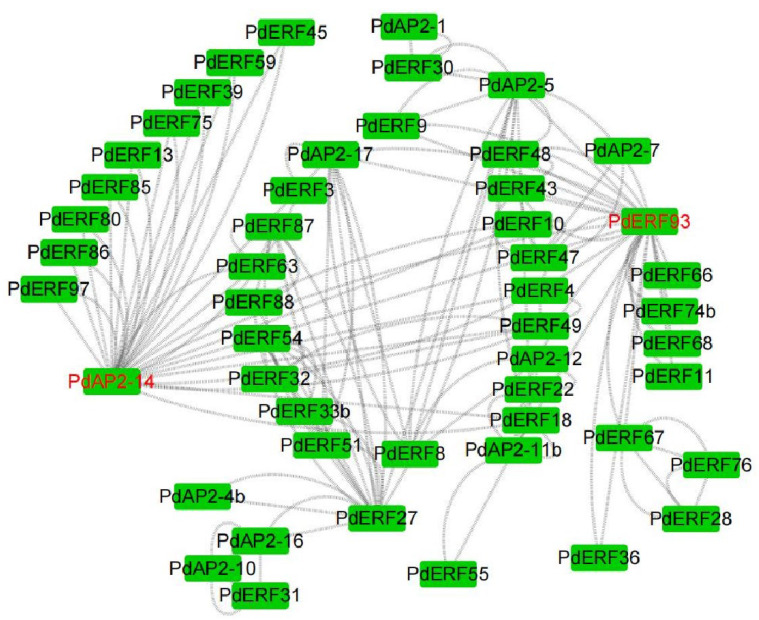
Protein–protein interaction network of PdAP2/ERF members. Red gene names represent central genes. The black dashed line represents the interaction between different proteins.

**Figure 11 biology-11-01520-f011:**
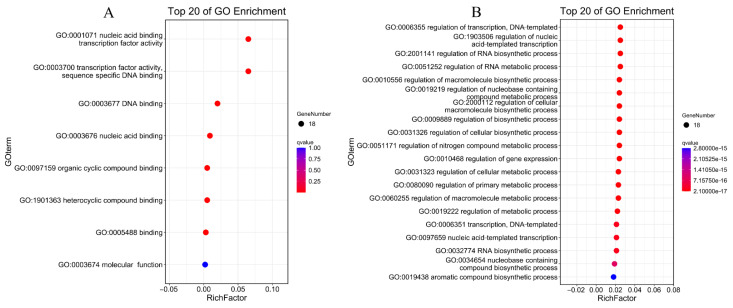
Top 20 significantly enriched GO terms for PdAP2/ERF-interacting proteins. (**A**): Molecular Function. (**B**): Biological Process.

**Figure 12 biology-11-01520-f012:**
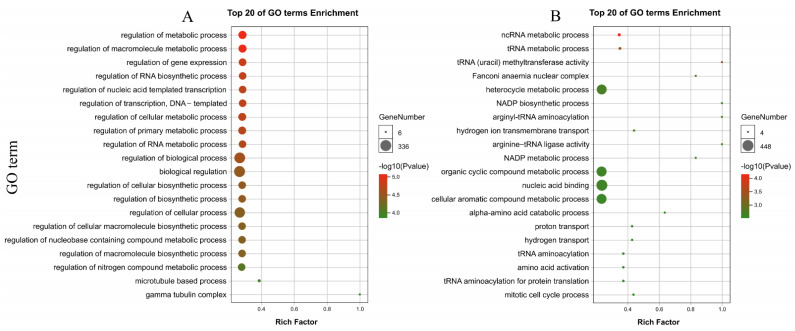
The top 20 significantly enriched GO terms for target genes. (**A**): Top 20 significantly enriched GO terms for AP2 target genes. (**B**): Top 20 significantly enriched GO terms for ERF/DREB target genes.

## Data Availability

The data presented in this study are available in the article.

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
