# Peer review of "Genome-Wide Analysis of the Almond AP2/ERF Superfamily and Its Functional Prediction during Dormancy in Response to Freezing Stress"

_biology, 2022, doi:10.3390/biology11101520_

Round 1
Reviewer 1 Report
AP2/ERF transcription factors are ubiquitous in plants, and these transcription factors play an important role in plant growth and development and in response to environmental stress. However,the main work of this paper is the bioinformatics analysis of the PdAP2/ERF gene family. Bioinformatic analysis of AP2/ERF has been studied many times in other species, so the authors need to further improve the innovation points of the article. I have some comments:
1. Line 50-51 “In summary, we conducted bioinformatics and expression pattern studies on the PdAP2/ERF genes”The bioinformatics analysis was performed on the PdAP2/ERF genes family,but gene expression pattern studies were conducted for 13 PdAP2/ERF genes.
2. Line 57、70 “biological and nonbiological stress”→biotic and abiotic stresses
3. Line 168-169 “ freezing stress (-5 (CK)”
The author needs to explain the background of the transcriptome data. Since -5℃is already freezing stress, why choose -5℃ as the control instead of low temperatures above zero served as CK?
4. Line 219 “homology alignment with Arabidopsis AP2/ERF members”
Arabidopsis and Almond species are so different, why not choose Malus domestica ,Prunus persica or PopulusL. for comparison?
5. Line 248 Figure 1 The author can choose Malus domestica,Prunus persica or PopulusL. for comparison, which would be more valuable.
6. Line 384-385 The authors found that the high expression of PdERF24 gene was 324 times at -30 °C. The authors should pay attention to whether there are similar reports of this gene in other species, which may be the key gene of Almond resistance to freezing stress.
Reviewer 2 Report
In this MS, the author analyzed all the AP2/ERF genes in almond, and their response to cold stress, providing useful information for stress response mechanism for trees. There are some suggestions for further modification:
1. For the title, “Genome-wide identification” is not appropriate. AP2/ERF genes are now totally new. They have been well studied in other plant species.
2. There are too much repetitive data. For phylogenetic analysis, figure 1 and figure 2 can be combined together. For chromosomal location, figure 3 , 4, 5 can be integrated.
3. The identification of target gene for AP2/ERF is important. Please also show the data for PdERF24, which showed significant cold response. To specify its target gene will outline a cold response pathway.
4. To analyze the expression of AP2/ERF genes in different kinds of almond, with different ability to tolerant cold stress, will give us more information about their function.
5. The words in figure 6, figure7, figure 8 and figure 9 are too small to read.
Round 2
Reviewer 2 Report
In this MS, the author analyzed all the AP2/ERF genes in almond, and their response to cold stress, providing useful information for stress response mechanism for trees. There are some suggestions for further modification:
1. For the title, “Genome-wide identification” should be modified. AP2/ERF genes are not totally new. They have been well studied in other plant species.
2. There are too much repetitive data. For phylogenetic analysis, figure 1 and figure 2 can be combined together. For chromosomal location, figure 3 , 4, 5 can be integrated.
3. The identification of target gene for AP2/ERF is important. Please also show the data for PdERF24, which showed significant cold response. To specify its target gene will outline a cold response pathway.
4. To analyze the expression of AP2/ERF genes in different kinds of almond, with different ability to tolerant cold stress, will give us more information about their function.
5. The words in figure 6, figure7, figure 8 and figure 9 are too small to read.
Author Response
Dear Reviewer,
On behalf of my co-authors, we thank you very much for giving us anopportunity to revise our manuscript.
We sought professional English editors to revise the article. Please check the PDF for the editing certificate.

Round 3
Reviewer 2 Report
In this revised MS, the author has made changes according to the suggestions, which make the whole story much better. There are still some minor things before publication:
1. For figures, the words in the figure are too small to read. Although the author has made some modifications, further improvement is still needed. For example, it is hard to see the numbers on the X-axis in Figure 2. The line for the error bars in Figure 9 is hardly seen. Also, some figure legend need to be modified. For example, there is no explanation for the red color in the chromosomes in Figure 3.
2. Although the AP2/ERF transcription factors are well known for their function in stress response, recently there are report of their role in grain yield regulation (Science, 2022, 377, 6604:eabi8455). Please also include this in the discussion part.
3. The expression level of the PdERF24 gene increased extremely at -30 °C, therefore, should be carefully analyzed in the following analysis. But please provide data for its target gene.
